# Chemical Diversity of Mediterranean Seagrasses Volatilome

**DOI:** 10.3390/metabo14120705

**Published:** 2024-12-14

**Authors:** Salomé Coquin, Elena Ormeno, Vanina Pasqualini, Briac Monnier, Gérald Culioli, Caroline Lecareux, Catherine Fernandez, Amélie Saunier

**Affiliations:** 1CNRS, Aix-Marseille University, Avignon University, IRD, UMR 7263 IMBE, 13397 Marseille, France; 2UMR CNRS SPE, UAR CNRS Stella Mare, Université de Corse, BP 52, 20250 Corte, Francemonnier_b@univ-corse.fr (B.M.)

**Keywords:** seagrasses, Mediterranean Sea, biogenic volatile organic compounds (BVOCs), chemical profile screening, HS-SPME/GC-MS

## Abstract

Background/Objectives: Biogenic volatile organic compounds (BVOCs), extensively studied in terrestrial plants with global emissions around 1 PgC yr^−1^, are also produced by marine organisms. However, benthic species, especially seagrasses, are understudied despite their global distribution (177,000–600,000 km^2^). This study aims to examine BVOC emissions from key Mediterranean seagrass species (*Cymodocea nodosa*, *Posidonia oceanica*, *Zostera noltei*, and *Zostera marina*) in marine and coastal lagoon environments. Methods: BVOCs were collected using headspace solid-phase microextraction (HS-SPME) using divinylbenzene/carboxen/polydimethylsiloxane (DVB/CAR/PDMS) fibers and analyzed by gas chromatography–mass spectrometry (GC-MS). Results: An important chemical diversity was found with a total of 92 volatile compounds (61 for *Z. noltei*, 59 for *C. nodosa*, 55 for *P. oceanica*, and 51 for *Z. marina*), from different biosynthetic pathways (e.g., terpenoids, benzenoids, and fatty acid derivatives) and with several types of chemical functions (e.g., alkanes, esters, aldehydes, and ketones) or heteroatoms (e.g., sulfur). No differences in chemical richness or diversity of compounds were observed between species. The four species shared 29 compounds enabling us to establish a specific chemical footprint for Mediterranean marine plants, including compounds like benzaldehyde, benzeneacetaldehyde, 8-heptadecene, heneicosane, heptadecane, nonadecane, octadecane, pentadecane, tetradecane, and tridecanal. PLS-DA and Heatmap show that the four species presented significantly different chemical profiles. The major compounds per species in relative abundance were isopropyl myristate for *C. nodosa* (25.6%), DMS for *P. oceanica* (39.3%), pentadecane for *Z. marina* (42.9%), and heptadecane for *Z. noltei* (46%). Conclusions: These results highlight the potential of BVOCs’ emission from seagrass ecosystems and reveal species-specific chemical markers.

## 1. Introduction

Volatile organic compounds (VOCs) are low-molecular compounds (<300 Da) characterized by low boiling points and high vapor pressures [1]. Biogenic VOCs (BVOCs), emitted by terrestrial plants, bacteria, algae, fungi, and animals, make up over 90% of global VOC emissions [2,3]. Terrestrial vegetation is the largest source, with estimated emissions exceeding 1 PgC yr^−1^ (10^15^ gC yr^−1^) [2,4]. Plant-emitted BVOCs are extremely diverse such as terpenoids (e.g., isoprene, monoterpenes, and sesquiterpenes), benzenoids and phenylpropanoids (e.g., benzaldehyde and cinnamyl alcohol), nitrogen and sulfur compounds (e.g., isothiocyanates and dimethyl sulfide called afterward DMS), and green leaf volatiles (GLVs) [5]. Each plant species has a unique BVOC profile or volatilome, influenced by its genome. For instance, Brassicaceae species emit isothiocyanates [6] rarely emitted by other plant families.

BVOC emission rates vary with biotic (e.g., herbivory and parasitism) and abiotic (e.g., light, temperature, and drought) factors [7,8] and act as defense compounds against these stresses [9,10,11]. Additionally, BVOCs play key roles in regulating interactions between species (e.g., plant insects) including pollination [12], herbivory [13], or response to pathogens [14].

BVOCs influence atmospheric chemistry by forming ozone (O_3_) and secondary organic aerosols (SOA) through reactions with oxidants like OH• and NO_3_• [15]. SOA influences atmospheric radiation by extending methane’s lifetime and acting as cloud condensation nuclei [15]. These processes affect air pollution, climate, ecosystems, and human health, emphasizing the importance of characterizing BVOC sources [16]. While BVOCs from terrestrial ecosystems have been extensively studied since the 1980s with over 2500 articles published since 1990 [17], marine environments remain comparatively understudied [18,19].

Oceans emit a large diversity of BVOCs including alkanes and alkenes [20,21], aromatics hydrocarbons (benzene, toluene, and xylenes) [22,23,24], and oxygenated VOCs (OVOCs, e.g., acetone and methanol) [25,26,27,28] as terrestrial vegetation. Oceans emit terpenoids, the largest class of specialized plant compounds [20,29,30,31], as well as BVOCs seldom or never documented in terrestrial environments, such as sulfur-containing compounds (e.g., DMS) [31,32,33,34,35], halogenated compounds [36,37,38] including brominated (e.g., CHBr_3_) and iodinated compounds (e.g., CH_3_I) [24,39]. These emissions, primarily highlighted in pelagic zones, are linked to sea surface processes [40], phytoplankton [41], and bacteria [19,42]. Recent reviews reveal limited research on benthic organisms, with 112 studies on macroalgae, 4 on seagrasses, and 2 on corals [19,43].

It can be speculated that BVOCs have a similar role in marine ecosystems to that in terrestrial environments. In the context of global change (rising temperature, ocean acidification, and salinity shifts [44]), studying BVOCs could reveal insights into ecosystem adaptation and resilience. Benthic ecosystems, covering vast coastal areas, may significantly influence the oceanic BVOC budget and atmospheric processes, as phytoplankton does [45,46]. However, emissions from benthic systems, particularly seagrass meadows, remain underexplored due to limited data [19,43]

Seagrasses, around 70 marine flowering plant species [47,48], form vast meadows that span over 300,000 km^2^ globally [49]. Seagrasses are an important component of marine and coastal lagoon ecosystems providing essential ecological services such as nursery habitat, food source, coastal protection, sediment stabilization [48,50], and carbon sequestration [51,52]. Despite representing less than 1% of total marine primary production, seagrasses are responsible for sequestering 10% of carbon in ocean sediments for millennia and mitigating the effects of climate change [53,54]. Nevertheless, these services are in decline since losses of seagrass meadows have been accelerating at an alarming rate of 110 km^2^ each year worldwide [55].

The Mediterranean region, one of the top biodiversity hotspots worldwide [56], had 468.54 km^2^ of seagrass meadows in 2016 [57]. Common species in the western Mediterranean Sea include *Posidonia oceanica* (L.) Delile, *Cymodocea nodosa* (Ucria) Asch., *Zostera marina* L., and *Zostera noltei* Hornem [58,59]. This region is also a hotspot of global change [60] with projected temperature increases of +0.81 to +3.71 °C by 2075–2100 in the upper layer (0–150 m) [44,61]. This could lead to reduced marine biomass, including fishery potential, in the western Mediterranean Sea [62]. Mediterranean seagrass meadows have declined by 13% to 50% since 1842 with the potential loss of 75% of their suitable habitats by 2050, and a risk of functional extinction by 2100 [63,64].

This project aims to address the gap in understanding seagrass BVOC production, potentially linked to their environmental responses with the screening of BVOC from the main Mediterranean seagrass species: *P. oceanica, C. nodosa, Z. marina*, and *Z. noltei.* The objectives are to (1) identify common compounds that define the chemical footprint of Mediterranean seagrasses, (2) determine species-specific chemical markers that characterize each plant’s unique chemical profile, and (3) explore the potential roles of these primary compounds in comparison to those found in terrestrial plants.

## 2. Materials and Methods

### 2.1. Plant Material

Four seagrass species classified as Least Concern on the IUCN Red List of Threatened Species were studied: (1) the endemic species, *Posidonia oceanica,* the most found throughout the Mediterranean basin except in the extreme south-east [65]; (2) *Cymodocea nodosa,* a thermophilic species extensively dispersed all over the Mediterranean basin, in the Sea of Marmara and in the Atlantic, which ranks second, after *P. oceanica*, in terms of occupied surface in the Mediterranean Sea [58]; (3) *Zostera marina,* the most widely distributed species, from the Atlantic Ocean to the Pacific Ocean, and from temperate regions to the Arctic Circle [66], which is mostly found as small isolated stands in the Mediterranean Sea and exists particularly in lagoons as a cold-water species; and (4) *Zostera noltei*, distributed across the coastal seas of Europe, which frequently grows in mixed meadows together with *C. nodosa* and/or *Z. marina* [65].

### 2.2. Study Sites

*C. nodosa, Z. noltei*, and *Z. marina* have been collected in the same meadow in Carteau Cove (south-east of France) (43°22′33.8″ N; 4°51′36.5″ E). This site is a shallow semi-enclosed soft-bottom zone located in the Gulf of Fos, about 35 km west of Marseille, and in the northwestern Mediterranean Sea [67]. *P. oceanica* has been sampled as close as possible next to the three first species (43°16′47.1″ N; 5°20′56.0″ E), about 40 km away, located in Marseille’s Bay. The sampling was performed on October 6th and 7th, 2023. In Carteau Cove, the water temperature was 14.5 °C and salinity was at 21.5 psu at a depth of less than 1 m. In Marseille’s Bay, the temperature was around 18 °C, salinity at 38 psu at a water depth of 2 m.

### 2.3. Seagrass Sampling

The permissions to request exemptions for protected species have been requested and granted by the competent authorities for these samples, in accordance with the French legislation. Species were collected and immediately placed in plastic bags containing seawater specific to each species, then transported in a cooler. In the laboratory, each species was placed in different 30 L tanks with synthetic seawater at the same conditions of salinity and temperature as in the field. A bubbling system was also installed to maintain water oxygenation.

### 2.4. Headspace Solid-Phase Microextraction (HS-SPME)

BVOC collection was performed within 24 h after seagrass harvesting. Before HS-SPME, each sample was taken out of the tanks, and each leaf was gently scraped with a scalpel to remove epiphytes while taking care to prevent leaf damage. Leaf samples were cut into small pieces and 1 g of fresh material was placed (separately for each individual) in 20 mL glass vials and hermetically sealed with PTFE/silicone septa. Based on the [68,69] method, the vials were maintained in a water bath at 50 °C for 10 min for equilibrium and the HS-SPME collection took place for 1 h. Collection of BVOCs from the headspace was carried out manually using a SPME holder and a previously conditioned StableFlex™ 1 cm divinylbenzene/carboxen/polydimethylsiloxane (DVB/CAR/PDMS) 50/30 μm fiber (Supelco Co., Bellefonte, PA, USA). Such a fiber allows to trap a large variety of volatile and semi-volatile compounds [70,71]. This extraction method, already used to collect BVOCs of seagrasses [72] and other marine organisms (e.g., macroalgae [73,74]), provides good sensitivity in short sampling times.

Blanks were performed using the same vials without plants. After sampling, the SPME fibers were stored at −20 °C before injection in GC-MS. BVOC collection was carried out in 5 replicates for each species.

### 2.5. Gas Chromatography–Mass Spectrometry (GC-MS) Analyses

Analyses were performed on a GC-MS instrument (7890B GC, Agilent Technologies^®^, Santa Clara, USA) equipped with an HP5-MS column (30 m × 0.25 mm × 0.25 μm; J&W Agilent Technologies^®^, Santa Clara, CA, USA) coupled to a mass spectrometer (MSD5977A, Agilent Technologies^®^). The HP-5MS column, commonly paired with SPME fibers in studies, is recommended for complex volatile mixtures due to its excellent resolution across various polarities and molecular weights. The GC-MS-temperature program has been adapted from [72]. Thermal desorption of the fibers was directly carried out to the GC column through the injector for 15 min at 250 °C in splitless mode. The gradient temperature was initially set at 70^◦^°C (2 min), then reached 200^◦^C at 3^◦^C min^−1^, and finally reached 315 °C at 15 °C min^−1^. Helium was used as a carrier gas with a constant flow of 1 mL min^−1^. The EI mode was operated at 70 eV and the mass range was 40–450 amu. The identification of VOCs was based on the comparison of their retention indices (RIs), determined using the retention times of a series of *n*-alkanes (C_8_ toC_20_), and on a spectral match with the NIST20 mass spectral libraries. To confirm the identity of certain terpenes, several standards were injected (β-cyclocitral, α- and β-citrals, geranylacetate, and *α*- and *β*-ionones). The relative abundance, that is, the relative percentage of each compound per sample, was calculated. The final table of all detected BVOCs, including relative abundance per compound and per species for each compound and classified according to the metabolomic pathways (benzenoid, fatty acid derivative, furanoid, sulfur-containing compound, terpenoid, and unknown), is available in Appendix A. Compounds only found in one sample (singleton compound) and all compounds detected in blanks were removed from the final table.

### 2.6. Statistical Analysis

Statistical analyses were performed with R (version 4.3.0) and Metaboanalyst 6.0 platform (https://new.metaboanalyst.ca/) accessed on 13 November 2024. Graphical representations were performed with ggplot2 and VennDiagram R packages.

Chemical richness was calculated as the total number of compounds per species. Chemical diversity considers both the richness and the relative abundance of each compound and was calculated using the Shannon–Weaver index and the vegan R package according to the formula:H′ = −∑ (p_i_ × ln(p_i_))
where p refers to the relative abundance of each compound [75,76,77]. An ANOVA test was applied to test statistical differences between species in terms of richness and chemical diversity.

A PLS-DA was performed to distinguish between species, and a heatmap based on an ANOVA statistical test was generated to reveal the 30 most discriminating compounds [78]. Both were produced, after a log_10_-transformation. These statistics illustrated the different chemical profiles and qualitative and semi-quantitative composition of BVOCs present in different species. A driven permutation Model Validation Analysis (MVA) (10^4^ perm., 6 components) using the RVAideMemoire R package [79] was applied to test if the chemical profile varied according to the species. The Conditional Error Rate (CER) was then calculated to ensure the robustness of the permutation model.

This study aimed to identify chemical markers specific to Mediterranean seagrasses, as well as species-specific chemical markers in order to develop a fingerprint as a tool to distinguish each species [80,81]. Several markers will be selected to form a chemical fingerprint. To define the chemical fingerprint of Mediterranean seagrasses, we selected relevant compounds produced by the four species and confirmed in the literature on Mediterranean seagrasses [72]. Chemical markers specific to each species must meet the following criteria: (1) The compounds must be highlighted in the heatmap and/or ranked among the top 15 compounds with the highest VIP scores from the PLS-DA, and (2) the compounds must be produced exclusively by the species (a species-specific compound) or, on average, at least two times more than other species. Both conditions must be satisfied.

## 3. Results

### 3.1. Volatilome Composition

In total, 92 compounds were detected: 61 for *Z. noltei*, 59 for *C. nodosa*, 55 for *P. oceanica,* and 51 for *Z. marina* (Appendix A). Each species presented specific BVOCs (9 for *C. nodosa*, 9 for *P. oceanica*, 5 for *Z. noltei,* and 2 for *Z. marina*) but also shared a large number of common volatile compounds (29, Figure 1). Among the 29 compounds are FAD (22 compounds including decanal, pentadecanal, pentadecane, 1-heptadecene, and isopropyl myristate), terpenoids (β-cyclocitral, β-ionone epoxide, β-ionone, farnesyl acetate, and geranyl acetone), and benzenoids (benzaldehyde and benzeneacetaldehyde). There were more chemical similarities between *Z. marina* and *Z. noltei* (10 compounds) and between *C. nodosa* and *P. oceanica* (8 compounds) than the other combinations.

### 3.2. Chemical Indexes

There were no differences between species in terms of chemical richness (ANOVA, *p* > 0.05; the average number of compounds: *Z. noltei*: 40.2, *C. nodosa*: 38.4, *P. oceanica*: 35.8, and *Z. marina*: 33.4) and diversity (ANOVA, *p* > 0.05; Shannon index: *P. oceanica*: 2.35, *Z. noltei*: 2.13, *C. nodosa*: 2.11, and *Z. marina*: 1.79) (Figure 2). The four species exhibited a similar number of compounds with similar abundances. For each species, the average value of the Shannon index (ranging from 1.79 to 2.35) indicated moderate diversity with some compounds being more abundant than others and each species having a different major compound.

### 3.3. Volatilome Classification

Detected compounds could be classified according to their biosynthetic pathways: fatty acid derivatives (FADs) (in relative abundance: *Z. marina*: 95.8%, *Z. noltei*: 85.3%, *C. nodosa*: 81.5%, and *P. oceanica*: 35.5%), terpenoids (*P. oceanica*: 15.9%, *Z. noltei*: 10.3%, *C. nodosa*: 6.6%, and *Z. marina*: 3.8%), and sulfur-containing compound (*P. oceanica*: 44.7%, *C. nodosa*: 11.0%, and *Z. noltei*: 3.2%, absent in *Z. marina*). A few compounds belonged to other biosynthetic pathways: benzenoids and furanoids as well as some unknown compounds (Appendix A). *C. nodosa* mainly emitted alkanes (38.8%), esters (32.8%), sulfur compounds (11%), alkenes (6.9%), aldehydes (5.7%), ketones (3.2%), ethers (<1%), alcohols (<1%), and some unknown compounds too (<1%) (Figure 3). *P. oceanica* mainly produced sulfur compounds (44.7%), aldehydes (15.9%), alkanes (14.0%), ketones (12.3%), esters (4.9%), ethers (3.4%), alkenes (2.0%), alcohols (<1%), and some unknown volatiles (<1%). *Z. marina* mainly released alkanes (79.4%), esters (11.0%), aldehydes (2.9%), ketones (2.7%), and, in low amounts (<2%), alkenes, ethers, alcohols, phenolic compounds, and unknown metabolites. *Z. noltei* mainly emitted alkanes (72.6%), aldehydes (6.7%), esters (6.4%), ketones (6.0%), sulfur compounds (3.2%), and, in low amounts (<2%), ethers, alkenes, alcohols, carboxylic acids, phenolic compounds, and unknown compounds. The major compounds per species in relative abundance were isopropyl myristate for *C. nodosa* (25.6%) (produced 10 to 37 times more than in the other three species), DMS for *P. oceanica* (39.3%) (produced 1.5 times more than in *C. nodosa* and 2 times more than in *Z. noltei*, while *Z. marina* produces no DMS), pentadecane for *Z. marina* (42.9%), and heptadecane for *Z. noltei* (46%) (both produced by the four species) (Appendix A).

### 3.4. Specific Volatilome

The four species presented significantly different chemical profiles (PLS-DA and permutation test; CER = 0.07, *p* < 0.001). According to PLS-DA (Appendix A), the chemical profile explained 39.7% of the differences between the four species with components 1 and 2. *C. nodosa* and *P. oceanica* are distinguished from other species along component 1, while component 2 separates the two *Zostera* species. The metabolites that most effectively explained species separation according to PLS-DA on components 1 and 2 were those with high VIP scores: citral, farnesan, neral, eicosane, and 3-ethyl-2-methyl-1,3-Hexadiene as key metabolites for *C. nodosa*; DMS, 2-pentylfuran, and pentadecanal for *P. oceanica*; nonanal, methyl stearidonate, and 3(E)-hexen-1-ol for *Z. marina*; and farnesyl acetate, 1-pentadecene, hexadecane, and isophorol for *Z. noltei* (Appendix A). The heatmap presented the discriminant compounds of the chemical profile across the four studied species. For example, the most specific compounds of *P. oceanica* were 1-tetradecene, α-ionone, alloaromadendrene, and diethyl phthalate. The biomarkers of *C. nodosa* were terpenes (e.g., citral, humulene, camphor, and neral) with eicosane and farnesan, while those of *Z. noltei* are isophorol, *β*-ionone epoxide, and dihydroactinolide (Figure 4). *Z. marina* has a specific chemical profile not because of specific compounds but because of it was the only species that did not produce DMS and produced fewer terpenes than the others. The heatmap highlights pentadecane as a biomarker. It produced alkanes with higher relative abundances compared to other species (e.g., tetradecane, pentadecane, and nonadecane). They were the only species to produce furanoids (2-pentylfuran (Appendix A).

The chemical markers of *C. nodosa* according to the heatmap and the PLS-DA’s VIP score were 3-ethyl-2-methyl-1,3-hexadiene, eicosane, produced respectively 2 and 15 times more than in the other species, and its species-specific compounds were camphor, citral, farnesan, humulene, and neral. The chemical markers of *P. oceanica* were α-ionone and pentadecanal, produced 11 and 6 times more than in the other species, and its species-specific compounds were alloaromadendrene, 1-tetradecene, and diethylphthalate. For *Z. marina*, the only chemical marker that matched our conditions was pentadecane, which produced 9.5 times more than in the other species. For *Z. noltei*, the chemical markers were farnesyl acetate, β-ionone epoxide, 1-pentadecene, dihydroactinolide, and hexadecane produced 2.2, 2.5, 2.8, 12, and 15.5 times, respectively, more than in the other species, and its species-specific compounds highlighted by the heatmap were isophorol.

## 4. Discussion

### 4.1. Volatilome Composition

Our results on volatilome composition highlighted that seagrass species can produce a large variety of volatile compounds belonging to different chemical families and bearing diverse chemical functions, as seen in other marine organisms [19,42,43]. Each of the four seagrass species studied in this work produced a similar number of compounds, ranging from 51 to 61 per species, slightly higher than previously reported for seagrass species. Earlier works reported 7, 16, 44, and 45 volatile compounds from *Z. noltei, P. oceanica*, and both *Z. marina*, respectively [72,82,83,84]. These differences are probably due to variations in extraction methodologies among studies. The number of compounds detected in our study was similar to those found in previous studies on macroalgae using the same collection fiber method [72,85]. The number of compounds can also be influenced by the sampling season, as BVOC emissions are known to vary according to season on both terrestrial plants [86] and other marine species [68,87]. This study focused on autumn volatilomes, as this season offers a standard production, free from summer heat stress and epiphytes, winter dormancy, and spring growth or flowering [88,89]. Monitoring throughout the year could reveal full chemical profiles and their seasonal variations.

### 4.2. Shared Compounds in Seagrass Species

The four species shared FAD, terpenoids, and benzenoids, which are common plant volatile compounds alongside phenylpropanoids and amino acid derivatives [90]. It is not surprising to find these types of compounds in seagrasses as well. Other marine organisms like macroalgae [91,92] such as coral [93] also produce these compounds.

Terpenoids are the largest class of plant-specialized metabolites [94] derived from isopentenyl diphosphate (IPP) and dimethylallyl diphosphate (DMAPP), synthesized via the mevalonic acid (MVA) in the cytosol and the methyl-erythritol-phosphate (MEP) in plastids pathways [95,96]. Shared terpenoids (e.g., β-cyclocitral and β-ionone) are apocarotenoids formed by carotenoid cleavage dioxygenases or non-enzymatic processes [97]. Apocarotenoids, more soluble and volatile than carotenoids, act as hormones, controlling their development, and signals regulating cell response to oxidative stress, serving critical biological functions [98,99]. Their role in the marine environment is unclear, but they likely offer protection against stressors, as seen in freshwater. β-cyclocitral acts as an inhibitor of competing microalgae during algal bloom in the eutrophic lakes [100,101] and increases with plant density, possibly as an inhibitor of competing organisms [102]. β-ionone and geranyl acetone produced by cyanobacteria and algae exhibit strong toxic effects on algae cell growth [103]. Cyanobacterial β-ionone and β-cyclocitral also trigger cell death in freshwater plants [104]. Both compounds certainly provide a similar defense against biotic stresses in the marine environment and may also contribute to protection against abiotic stresses. In freshwater, the conditions of high light and temperature can promote the emission of β-ionone [105] and β-cyclocitral [106]. The four species’ volatilomes also shared an oxygenated sesquiterpene: Farnesyl acetate commonly produced by brown algae [107], but its role in marine areas remains unknown. In terrestrial plants such as *Abelmoschus moschatus*, farnesol acetate is produced in seeds where it shows an antibacterial activity [108]. It is also an important sex pheromone for the Click beetle, attracting females [109]. In the marine environment, it could potentially have an antibacterial role as well as being an important signaling agent.

Benzenoids derived from L-phenylalanine constitute a large class of structurally diverse volatile compounds involved in plant reproduction and defense [94]. Benzaldehyde and benzenacetaldehyde have already been found in *P. oceanica* and seaweeds [68,72,87,110]. The role of benzenoids in marine environments is unknown. In the terrestrial environment, benzaldehyde derivatives show growth inhibition and antioxidant activities on *Brassica campestris* [111], and most floral volatiles include benzenoid compounds own to their high smell potential required to attract pollinators [112,113]. For example, once pollinated, *Petunia* flowers produce ethylene, which rapidly induces the down-regulation of biosynthesis and emission of all benzenoids [113].

FADs derive from linoleic and linolenic acid by deoxygenation catalyzed by lipoxygenases (LOX) [114]. Some of them are important herbivore-induced plant volatiles (HIPVs) immediately released after insect damage [94]. Although their exact roles in marine environments remain unknown, marine organisms produce these compounds in significant quantities (e.g., macroalgae [91,115,116] and seagrasses [43,84]).

### 4.3. Major Compounds Released by the Four Studied Species

For each species, the Shannon index average indicated specific major compounds for each species. For *C. nodosa*, the main compound was isopropyl myristate, already highlighted in BVOC produced by red algae (*Corallina elongata* and *Polysiphonia denudate* [115,117]), brown algae (*Ulva punctaria* [117]; *Cystoseira compressa* [118]), and green algae (*Codium tomentosum* [119]).

DMS was the main compound of *P. oceanica* as shown by [72]. DMS is the most emitted sulfur compound by the oceans to the marine troposphere [33,35] with annual emissions exceeding 107 t. It is a key component of the ocean sulfur cycle and has a global role in atmosphere–ocean feedback processes [120,121]. DMS acts as a defense compound against herbivory, released in higher amounts during grazing on macroalgae, which reduces feeding and limits macroalgae consumption [119,122].

The two *Zostera* species mainly produced alkanes: pentadecane for *Z. marina* and heptadecane for *Z. noltei*. Pentadecane is commonly produced by brown macroalgae [123], while heptadecane has been mainly described in red macroalgae [68] with varying emissions according to the season [68,123]. These alkanes decrease in response to heat shock [124]. Our results showed that the major compounds produced are FADs, particularly for *Z. marina*. These compounds could be plant membrane degradation products, especially as the analytical method used here required heating to 50 °C. This method is similar to those in studies on benthic organisms (e.g., macroalgae [69,116,119]). The optimal SPME temperature range is between 40 °C and 70 °C [125,126], and many studies indicate that heat does not alter BVOC chemical profiles but primarily affects the abundance of highly volatile compounds [126,127]. Moreover, the method used in the present study and previous ones [69,72,73] requires plants to be cut, thus inducing GLV production. GLVs are C6 aldehydes, alcohols, and esters forming a distinctive scent when leaves are damaged [128]. The only two GLVs recorded in this study are 3-hexenol in the two *Zostera* species and hexanal in *Z. noltei* and *C. nodosa* [129], suggesting that *Zostera* spp. and *C. nodosa* produce defense compounds or at least a reaction to herbivory. For *P. oceanica*, no GLV was identified in this study.

### 4.4. Specific Volatile Markers by Species

This study revealed significant interspecific variability in the BVOC profile, likely due to species-specific phenotypes, as site variability was limited. Each species had its own volatilome as a marker of plant genotype [2,130,131].

*P. oceanica* was discriminated by α-ionone, alloaromadendrene, and diethyl phthalate. α-ionones may have roles similar to its isomer β-ionone, which possesses numerous ecological roles (as previously discussed). Alloaromadendrene contributes to oxidative stress resistance in terrestrial plants [132], suggesting these two volatiles could act as signal molecules against abiotic stresses in *P. oceanica*, although further studies are needed to assess such an ecological role. Diethyl phthalate, often described as synthetic, can have a biogenic origin, produced by certain bacteria species [133,134], a cyanobacteria [135], and some terrestrial plants (e.g., fescue [136] and leek [137]).

The most discriminating compounds for *C. nodosa* were terpenoids (camphor, citral, humulene, and neral). Since the 1970s, various terpenes have been demonstrated to be toxins, repellents, or attractants, with ecological roles in interactions among organisms [138]. Terpenes may also act as defense compounds for many marine organisms, including algae, sponges, corals, molluscs, and fishes [139] helping sedentary marine organisms to prevent colonization by epiphytes [140].

## 5. Conclusions

This study highlighted that Mediterranean seagrass species produce a wide variety of BVOCs (92 compounds). Common BVOCs, usually found in terrestrial plants, were detected (e.g., β-ionone, citral, and camphor) along with compounds specific to marine environments (e.g., DMS and β-cyclocitral). Each species presented shared BVOCs (e.g., benzaldehyde, geranylacetone, and tridecanal) as well as specific markers such as α-ionone or alloaromadendrene for *P. oceanica* and various terpenoids for *C. nodosa*. We can expect that marine BVOCs could serve similar roles to those in terrestrial ecosystems such as offering protection to seagrass meadows against climate stress (among others), though further investigations are needed to confirm this hypothesis.

## Figures and Tables

**Figure 1 metabolites-14-00705-f001:**
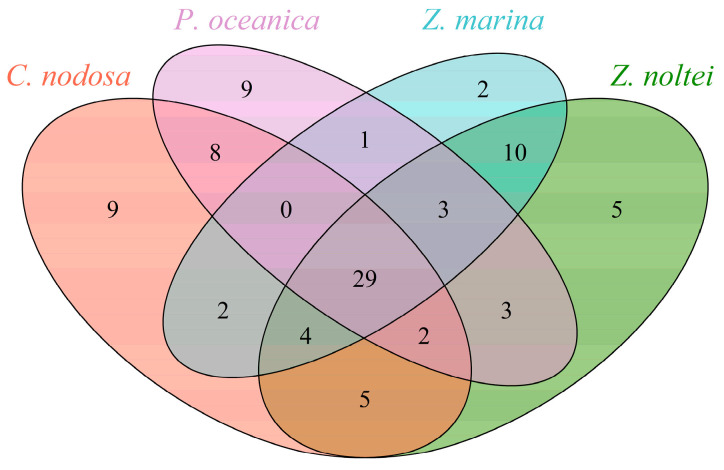
Venn diagram analysis of the number of shared and unique volatile compounds between the four Mediterranean seagrasses in this work. Finally, the chemical markers of Mediterranean seagrasses are benzenoids, such as benzaldehyde and benzeneacetaldehyde, and FADs, such as 8-heptadecene, heneicosane, heptadecane, nonadecane, octadecane, pentadecane, tetradecane, and tridecanal.

**Figure 2 metabolites-14-00705-f002:**
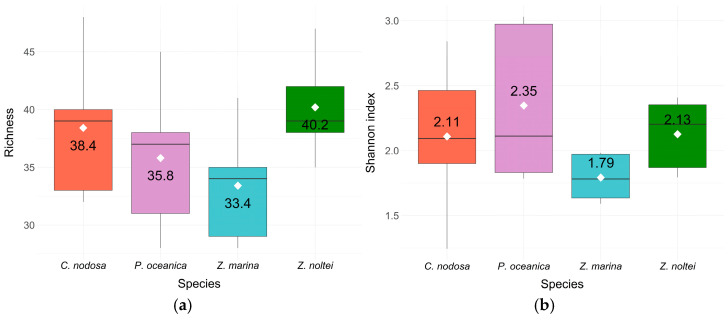
Chemoecological indexes: richness (**a**) and Shannon–Wiener diversity (**b**) of volatile compounds produced by the four main Mediterranean seagrasses. The white squares with the value indicate the average index. An ANOVA was performed to highlight differences between species and no significant variations were detected (*p*-value > 0.05).

**Figure 3 metabolites-14-00705-f003:**
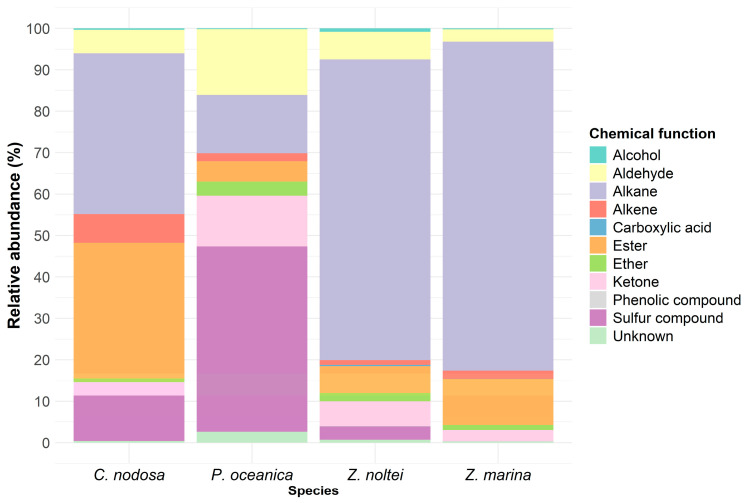
Composition by the chemical function of the volatilome of the four Mediterranean seagrasses.

**Figure 4 metabolites-14-00705-f004:**
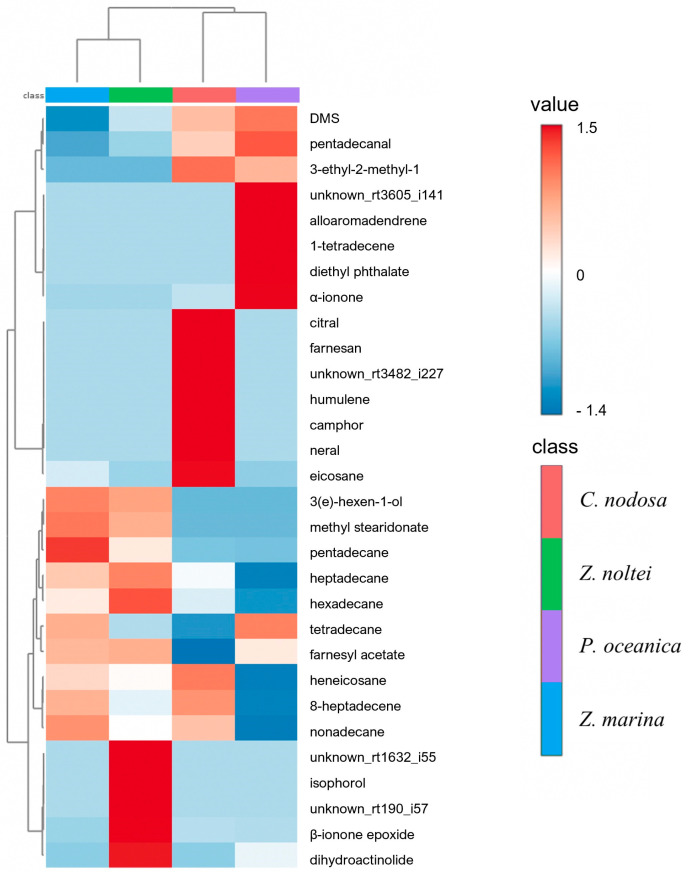
Heatmap based on the 30 most discriminating compounds (*t*-test/ANOVA) of the volatilome of the four seagrass species studied in this work.

## Data Availability

The original contributions presented in this study are included in the article/Appendix A. Further inquiries can be directed to the corresponding authors.

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
