# Peer review of "Chemical Diversity of Mediterranean Seagrasses Volatilome"

_metabolites, 2024, doi:10.3390/metabo14120705_

Round 1

Reviewer 1 Report

Comments and Suggestions for Authors

The manuscript titled “Chemical diversity of Mediterranean seagrasses volatilome” deals with the volatile organic compounds emissions from key Mediterranean seagrass species (Cymodocea nodosa, Posidonia oceanica, Zostera noltei, and Zostera marina) in marine and coastal lagoon environments using HS-SPME/GC-MS.  The subject is interesting and presents some novelty. However, there are some weak points in the research, concerning overall design of the experiment and preparation of nanocomposite The article was not found suitable for publication in Metabolites the present form, it needs major revisions.

I would like to give further suggestion on the following matters:

Line 159: No clear data are reported concerning about the conditions used for SPME analysis, concerning time of equilibrium and its choice: were different trails carry out? Were trials finalised to assess time of equilibrium and analytes take off carried out? What about the linearity and repeatability applied analytical techniques? Please explain SPME analysis in more detail.

Line 166:  Add the lenght of SPME fiber, 1 or 2 cm.

Line 167:  Add a city and country name for (Supelco Co)

Line 176: Why the authors used only one column (HP-5MS fused-silica capillary column) for the studied sample. In most cases some volatile compounds, which co-elute on one column can be distinguished easily on the other column with different polarity. It is much better to work two different columns for volatile studies.

Lines 178-378: Jerkovich was changed as Jerkovic

Improve the resolution of Figures.

Author Response

Line 159: No clear data are reported concerning about the conditions used for SPME analysis, concerning time of equilibrium and its choice: were different trails carry out? Were trials finalised to assess time of equilibrium and analytes take off carried out? What about the linearity and repeatability applied analytical techniques? Please explain SPME analysis in more detail.

  • Time of equilibrium is important effectively. To develop our method, we did not conduct specific tests, we are based on studies that tested various sampling times on similar organisms (e.g., Radman 2022 and 2023). Following their recommendations, we extended sampling slightly (1h) to maximize compound detection without encountering saturation. We added these details in the Section 2.4, line 137 to justify our method.

Line 166:  Add the length of SPME fiber, 1 or 2 cm.

  • We added the length of the fiber to line 141 and some additional information “StableFlex™ 1 cm divinylbenzene/carboxen/polydimethylsiloxane (DVB/CAR/PDMS) 50-30 um fiber”

Line 167:  Add a city and country name for (Supelco Co)

  • We added the city to line 142.

Line 176: Why the authors used only one column (HP-5MS fused-silica capillary column) for the studied sample. In most cases some volatile compounds, which co-elute on one column can be distinguished easily on the other column with different polarity. It is much better to work two different columns for volatile studies.

  • Thank you for pointing this out. We have added in the Materials and methods an explanation of this choice. In the section 2.5, lines 152-155 “The HP-5MS column, commonly paired with SPME fibers in studies, is recommended for studying complex volatile mixtures due to its excellent resolution across various polarities and molecular weights.” Certainly, other columns may allow to detect additional compounds, but SPME fibers may not capture them effectively for the lightest compounds.

Lines 178-378: Jerkovich was changed as Jerkovic

  • Thank you for the correction of the text. References have been corrected (line 155 and line 359 now).

Improve the resolution of Figures.

We agree with this comment for the Heatmap figure. The picture has been modified. All other figures have been transmitted in high resolution

Reviewer 2 Report

Comments and Suggestions for Authors

The article is devoted to the study of the composition of biogenic organic volatile compounds emitted by seagrasses. The article contains all the necessary sections. Publication of this article in the journal Metabolites is possible after elimination of minor comments.

The Introduction and Discussion sections are very long and contain redundant information. Perhaps they should be shortened. In the Introduction section, lines 34-67 could be shortened. Discussion section Lines 322-327 repeat information from the Results section.  This repetition should be eliminated.

 The Conclusions section should be shortened and only the specific conclusion of the paper should be described. Literature citations should not be present in this section

Literature citations are incorrectly organized - line 73 and line 388.

Table S1: Table of volatile organic compounds by biosynthetic pathway and relative abundance (± Standard deviation) in the four Mediterranean seagrass species extracted by HS-SPME and analyzed by GC-MS should be added

Author Response

The Introduction and Discussion sections are very long and contain redundant information. Perhaps they should be shortened. In the Introduction section, lines 34-67 could be shortened.

  • As requested,we have shortened the introduction and discussion by removing redundant or non-essential information to enhance clarity and focus. Additionally, we revised several sentences for conciseness. All changes are documented in the “revision tracking” file. The introduction was reduced from 92 lines (1101 words) to 66 lines (749 words) (a reduction of 32%), and the discussion from 133 lines (1572 words) to 104 lines (1225 words) (a reduction of 22%). These revisions have reduced the manuscript by one page.

Discussion section Lines 322-327 repeat information from the Results section.  This repetition should be eliminated.

  • The repetitions have been eliminated and one sentence has been added to the result section.

 The Conclusions section should be shortened and only the specific conclusion of the paper should be described. Literature citations should not be present in this section

  • Citations have been removed from the conclusion and the information added to the discussion. Only the main results are now included in the conclusion.

Literature citations are incorrectly organized - line 73 and line 388.

  • References have been corrected on line 155 and 359

Table S1: Table of volatile organic compounds by biosynthetic pathway and relative abundance (± Standard deviation) in the four Mediterranean seagrass species extracted by HS-SPME and analyzed by GC-MS should be added

  • Thank you for pointing this out. It was an omission and has been transferred with the other Supplementary files.

Reviewer 3 Report

Comments and Suggestions for Authors

This article concerns the study of BVOC emission of biogenic volatile compounds (volatilome) from Mediterranean seagrasses Cymodocea nodosa, Posidonia oceanica, Zostera noltei, and Zostera marina. The authors used headspace solid phase microextraction (HS-SPME) on divinylbenzene/carboxen/polydimethylsiloxane (DVB/CAR/PDMS) fibers and standard GLC-MS procedures. The substances were identified by specific retention times, molecular masses, comparing the fragmentations in the mass-spectra with the corresponding databases and sometimes by the using of authentic standard samples. The use of analytical methods seems to be adequate.

The results and their analysis are very impressive. The authors totally have isolated totally 92 compounds from four species. They estimated specific compounds for each species and overlapping compounds, their distribution by biosynthetic classes, chemical functions etc. The used all necessary statistical procedures. These results really highlight the potential of biogenic volatile compounds emission from seagrasses and reveal species-specific chemical markers.

The article is well written and interesting for the readers but has several imperfections, both minor and serious ones.

I recommend to deсode PgC yr⁻1 abbreviation, to decode the DMS (dimethyl sulfide) (Line 75), to change the article “a” with article “an” before “unique (Line 75).

The Introduction seems to be too large and excessive. Attempt, please, to shrink it by three or four folds.

Author Response

I recommend to deсode PgC yr⁻1 abbreviation, to decode the DMS (dimethyl sulfide) (Line 75), to change the article “a” with article “an” before “unique (Line 75).

  • We deсoded PgC yr⁻1 in line 38 “(1015 gC yr⁻¹) “. We also changed the article “a” to “an” in line 42 and we explained the DMS abbreviation in line 41.

The Introduction seems to be too large and excessive. Attempt, please, to shrink it by three or four folds.

As requested,we have shortened the introduction and discussion by removing redundant or non-essential information to enhance clarity and focus. Additionally, we revised several sentences for conciseness. All changes are documented in the “revision tracking” file. The introduction was reduced from 92 lines (1101 words) to 66 lines (749 words) (a reduction of 32%), and the discussion from 133 lines (1572 words) to 104 lines (1225 words) (a reduction of 22%). These revisions have reduced the manuscript by one page.

Round 2

Reviewer 1 Report

Comments and Suggestions for Authors

I reviewed the article titled “Chemical diversity of Mediterranean seagrasses volatilome" again as a referee. The researchers have made necessary corrections. Therefore, I can recommend the manuscript for publication in the Metabolites.